# Evaluation on Early Drought Warning System in the Jinghui Channel Irrigation Area

**DOI:** 10.3390/ijerph17010374

**Published:** 2020-01-06

**Authors:** Shibao Lu, Yizi Shang, Hongbo Zhang

**Affiliations:** 1School of Public Administration, Zhejiang University of Finance and Economics, Hangzhou 310018, China; lushibao@zufe.edu.cn; 2State Key Laboratory of Simulation and Regulation of Water Cycle in River Basin, China Institute of Water Resources and Hydropower Research, Beijing 100101, China; 3School of Environmental Science and Engineering, Chang’an University, Xi’an 710054, China

**Keywords:** precipitation, soil moisture, drought index, early droughts warning and drought assessment

## Abstract

With the economic growth, continuous global environment deterioration, and increasingly serious water resources shortage, droughts have become more and more serious and produced great impacts on both the regional ecology and sustainable economic development. This paper has established the “green, blue, yellow, orange, and red lights” as the early warning grades for agricultural droughts. By using the two influencing factors, namely precipitation and soil moisture, this paper has established the drought assessment index evaluation model using weighted coupling method. It has carried out the analogue simulation of the early drought warning based on the Jinghui Channel’s 2013 water source situations. The soil moisture in January and February is relatively ideal, and the actual early drought warning is expressed by the “green light”. The soil moisture deficit is comparatively serious in March, but the situation concerning water inflow is ideal with the “green light”. Actually, the early warning signal is basically consistent with the soil moisture drought degree between April and August. The actual early warning is expressed by the “green light” as well, but the soil moisture is not so ideal, however, this is the seeding time of the winter wheat so the lack of soil moisture has no impact on the crops output. In November and December, the winter wheat is at the growth and development stage and does not need much moisture. At this stage, the soil moisture is relatively poor. By integrating the time effects into the early drought warning system, this paper provides administrators of irrigation areas with a scientific decision-making based on the drought control measures.

## 1. Introduction

Drought is a natural disaster that has the widest impact and causes the largest economic loss to agriculture worldwide [1,2,3]. Research shows that losses caused by drought in the world have accounted for 15% of total losses caused by natural disasters. It has deeply affected the development of human life and social economy by also posing serious threats to the natural environment. According to the UENP estimation, 35% of the land and 20% of the world population are affected by droughts and desertification, where the annual desertification area can reach a maximum of 6 million hectares. These two disasters have caused losses as much as approximately 26 billion US$ on agricultural production. According to the LANBURTON research, the annual average of economic losses caused by natural disasters in the world is 40 billion US$, and that of those caused by drought accounts for 15% (6 billion US$, approximately). 

Throughout the world, drought impacts are increasing daily [4,5,6]. Mankind has been mindful of the fact that water shortage caused by droughts can only be avoided or mitigated to some extent, but it is inevitable. An increasing number of people have attached great importance to the preparations prior to drought and timely focused on the course in connection with its occurrence, development, extinction, and fading away for the purpose of taking effective and time-based measures to mitigate or relieve its impact losses.

A recent study looked at evaluation, management, forecasting, and early drought risks warning [7,8,9]. Sharvelle et al. [10] has put forward the urban droughts and water shortage management mode based on the risk evaluation and management theory: 

The research on drought forecasting and management has also attracted the American government’s attention to natural disasters. All sectors of the society have put more and more emphasis on prevention, especially the federal government, which has attached more importance to their mitigation and risk management. Important constituent parts of the U.S. drought policy are risk-based management and prevention schemes [11]. Yan et al. [12] has systematically researched based on the theoretical drought risk evaluation and management and has also put forward the corresponding plans and preventive actions, which are based on 10 steps. 

In terms of the research and services concerning the early drought warning, this paper is carried out by mainly adopting the statistical model, Markov chain transition probability, and by employing the precipitation-related factors, Palmer drought, and standardized precipitation indexes [13,14,15,16]. Bagirov [17] has established a multiple linear regression model using the sowing delay date, monthly precipitation and rainy days, and so forth deriving from the precipitation data in order to estimate the one month output before harvesting and has brought about a final early warning model to estimate the crop output at the time when crops are about to be harvested. The models have been further optimized and improved. In 2009, Mallick [18] established a stepwise regression formula by using the cumulative soil moisture index (CSMI) in the growing season, precipitation days in August, and the delaying sowing time. The improved model reduces the absolute output estimation error to 13.7% from 18.5%. The author considers that the soil moisture index and other precipitation-related variables can be used to develop other early warning models in the drought areas.

Bonaccorso [19] calculated the standardized precipitation index by means of the precipitation data in 1968 and forecasted the time and drought grade and its variation time by using the uniform and non-uniform Markov chain transition probability models. The short-term verification results showed that the Markov chain could nicely forecast droughts at the monthly scale, and when it was non-uniform, it would have better forecasting effects. It could distinguish the situations of the initial month and fully reflect the climate impacts on drought, especially on the seasonal precipitation. Rayne [20] et al. carried out the early drought warning in Virginia using the drought index PDSI and the non-uniform Markov chain transition probability, which provided references for decision-makers. Hatmoko [21] mainly researched on the method of using the satellite remote sensing means to explore the early drought warning. He considered that continental droughts were mainly associated with the southwest monsoon, a warm cyclone blowing from Southwest Asia, El Nino, and the Southern Oscillation, and the profiles of elements in relation to the precipitation distribution, such as the sea surface temperature, snow cover, cloud cover, wind speed/direction, and atmospheric temperature/humidity could be monitored by satellites. The monitoring results of these elements across the world and in some regions provided by the low earth orbit satellites are helpful to establish an ocean–atmosphere model to estimate the monsoon development and provide the early warning for the drought caused by monsoons.

The drought indicators and assessments. Gommes proposed the national rainfall index (RI) in 1994, which currently is often used in studies comparing continental-type climate scales and abnormal rainfall patterns. In 2004, Gonzalez established an index of drought with the application of the main recurrence frequency. This index was applied to the Ibero Peninsula (Brodo, TX, USA). In addition, there are many drought evaluation indicators based on satellite and aerial remote sensing, such as vegetation state index (VCI), crop water shortness index (CWSI), normalized water index (NDWI), and surface vegetation drought index (TVDI).

Haro-Monteagudo et al. [22] has established an early drought-warning index system, with the indexes mainly including control station flow, average surface precipitation, the crop comprehensive water shortage, and the grain’s estimated yield reduction rates. It is feasible to use the early drought-warning system which focuses on the risk techniques for the early agricultural droughts warning. The system can analyze the droughts probabilities at different degrees in each period. Passing through the use of Markov chain method, we can get the stability of the drought transition probability [23]. Therefore, probability in the next period can be found in the drought probability transition matrix based on the drought situations of this period. Alcántara-Ayala [24] has researched on each natural disaster’s inducing mechanism and process, established a monitoring, evaluation, and early warning system, and studied the disaster-forming conditions and general inducing models.

Using the method of monitoring agricultural drought by remote sensing, Fan [25] et al. considered the forecasted precipitation and maximum air temperature in the future early warning period on the basis of the soil moisture remote sensing monitoring and established an empirical model to convert those above-considered factors into the soil moisture correction values for the early drought warning.

The management of the drought. Limited by drought indicators and forecasting methods, early management of drought was mostly based on the historical drought data, including drought data, disaster loss data, and related meteorological, hydrological, and monitoring data. With the deepening of the research on the early warning mechanism of the indicator system, the original decision support system (DDS), which used the “monitoring, evaluation, early-warning” as its management framework, began to appear in the field of meteorology and agriculture [26,27,28,29]. There are still problems such as inability to provide relevant information about water supply and insufficient decision-making options. The emergence of an integrated water resources management system made up the shortcomings of DDS. For example, Merabente and others developed a water resources decision support system in Fukuoka, Japan, which is able to assess the water supply systems to different drought events and help the decision-makers: the best water supply plan [30,31]. Pallottino developed a decision support system in 2005, which can generate decision schemes in multihydrological scenarios, providing more risk information for decision-makers to make decision schemes [32,33,34]. In 2012, Won et al. proposed an agricultural drought decision support system based on risk assessment, which carried out assessment and early warning using the reservoir drought indicators and soil water content indicators and can make risk decisions based on hydrological frequency, climate change, and historical drought condition [35,36].

Judging from the current worldwide development of drought warning and management, attentions are increasingly turning to the area of reducing the risk of drought, which means the efforts of reducing drought are through various disaster mitigation actions. At the disaster level, drought management has also shifted from crisis management in the study of temporary emergency measures after the occurrence of drought, to the development of a drought plan or plan before the occurrence of drought, so that when a drought occurs, corresponding drought plan measures can be implemented to prevent drought and reduce disasters based on the characteristics of drought risk [37].

Targeting shortcomings of past studies, this paper has proposed to select two influencing factors, precipitation and soil moisture, in order to establish the drought evaluation index (D) in the form of weighted coupling and establish the “green, blue, yellow, orange, and red lights” as the early warning grades for agricultural drought through the determination of the water source situation index (S). This method uses the above two indexes of the irrigation area, (D) and (S), to reflect a comprehensive index value concerning the disaster crisis that the irrigation area may face in the future. The method combined the water source situation, namely the supply–demand relationship, to represent future possible drought risks. It is an important breakthrough in the research on early drought warning systems.

## 2. Research Method and Study Area

### 2.1. Study Area

Jinghui canal: the Zhengguo canal was built during the warring states period in 246 Before Christ. After the drought in Guanzhong in the late 1920s, the irrigation project of Jinghui canal was built with a planned irrigation area of 105,437 acres at that time, and a real irrigation area of 82,372 cres by 1949. Since then, the canal head project and irrigation and drainage system have been expanded and rebuilt. The irrigation area is irrigated by a single source of water and was developed into a multisource irrigation system combined with canal wells. Taking Jinghui Canal as the center, connecting Baoji Gorge, Wool Bay, Fengjiashan, and other irrigation areas in the west, Dongfanghong, Luohui Canal, and Lei Yanghuang irrigation areas in the east, Baoji in the west, and Weibei Plateau in the hundreds of miles from the Yellow River in the east, a river canal was formed into a network, where the surface water is combined with groundwater, and the large-scale irrigation system of nearly 670,000 hectares of the tank land was turned into the main grain production base in Shaanxi Province(see Figure 1).

### 2.2. Establishment of an Early Drought Warning Index System

#### 2.2.1. Determination of Drought index (D)

This paper has introduced the process of establishing and evaluating the drought index (D) of irrigation areas by using the fuzzy synthetic evaluation method [38,39]. This paper mainly has selected two influencing factors, which are precipitation and soil moisture, to establish the drought evaluation index (D) in the form of weighted coupling, with particulars given as follows:(1)Precipitation

In regards to the arid and semi-arid regions, the drought occurrence possibility is mainly dependent on precipitation in the meteorological conditions, so the precipitation index is taken as an evaluation one. In the early drought warning, the standardized precipitation index (SPI) is simpler than the Palmer drought severity index and is well-adapted, so it is used to represent meteorological drought in the irrigation area.

(2)Soil moisture

With respect to the Jinghui Channel Irrigation Area, the soil moisture is a direct drought representation, so its impacts are taken into consideration during drought evaluation. Also, its statistics are picked out from soil moisture data which are found in the official monitoring results provided by Jinghui Channel Management Office.

(3)Classification of Drought grades

The classification of drought grades should fully reflect the drought changes scope in the research area. According to provisions in the “Compilation Guidelines for Drought Emergency Plan”, the drought situations in the Jinghui Channel Irrigation Area are classified into the following five grades: no drought (V1), mild drought (V2), moderate drought (V2), serious drought (V4), and extreme drought (V5).

According to the comprehensive analysis and research and in combination with actual situations of the irrigation area, the indexes are classified, with specific results shown in Table 1, as follows:

The comprehensive drought evaluation index is formed through the coupling of the standardized precipitation index and soil moisture. The coupling means that there is a fuzzy synthetic evaluation method. The two indexes are subjected to fuzzy evaluation in order to get the drought index (D) grade. The precipitation to soil moisture ratio is 0.45:0.55.

#### 2.2.2. Water Source Situation Index (S) Determination

The water source situation index mainly represents the system’s demand–supply equilibrium in future periods [40,41,42]. The research shows that when the drought occurs in the irrigation areas, which is associated with severity, what is more important is that the insufficient water supply at different developing stages of the crops is more likely to cause the drought events. What happens in most cases is that the actual situation does not have signs of droughts, but droughts may happen because crops have great demands for water, and the local precipitation and water supply cannot satisfy such demands. Therefore, the water source situation impacts on occurrence and development must be taken into consideration in the early drought warning.

Jinghui Channel Irrigation Area is a large irrigation area (II) that has adopted the channel’s well-combined irrigation method. In the irrigation area’s water supply system, the water channels and water wells have played significant roles. Moreover in recent years, long-term operation has resulted in senescence and disrepair of the engineering facilities; the water delivery is rough, its supply from headwaters of the area is insufficient, and over-exploitation of groundwater has led to the restricted water supply capacity. Therefore, the author has comprehensively considered the channel head diversion and the amount of the area’s groundwater mining at each stage as the bases for the water source situation index calculation.

In order to reflect the supply–demand relation in terms of the water demands for the irrigation area, this paper has selected the demand at each stage in the year of research according to the irrigation programs of different hydrologic years and designed the specific expression mode of the water source situation index. Monthly frequency analysis of 21 years data (between 1993 and 2013) has been carried out, and estimation of possible water inflow into the irrigation area has also been conducted according to different empirical frequencies (P (Q ≥ Q_P_)). The balance between the estimated water inflow and the water demand at the corresponding period is taken as the water deficit in such period, and the percentage ratio between the water deficit and water demand is the water shortage rate. Based on Huang Wenzheng’s analysis described in the Establishment and Application of Early Drought Warning System for Reservoirs, we get the water source situation analysis grade table (Table 2) as follows.

#### 2.2.3. Establishment of Early Drought Warning Framework System

##### Determination of Early Warning Light Signals

Entropy is a measure of the system state’s uncertainty, and the future water shortage degree in the irrigation area is uncertain, so the entropy should be used in this paper to determine the number of early warning signals for the water shortage degree.

Assuming that there are *n*_0_ possible states for the random event *x*, the occurrence probability of each state is $p_{i}(i=1,2,\ldots n_{0})$, and the entropy H(x) for the event x is
(1)$$H(x)=-\sum_{i=1}^{n_{0}}p_{i}\;\log_{2}(p_{i})).$$

When the system has an equal probability $p_{i}=\frac{1}{n_{0}}$; put it into the Formula (1), and H(x) will be expressed by
(2)$$H(x,y)=-\sum_{i=1}^{n}\frac{1}{n_{0}}\log_{2}(\frac{1}{n_{0}})=\log_{2}(n_{0})$$

With respects to the relevant events x and y, the uncertainty can be expressed as H(x,y):(3)$$H(x,y)=-\sum_{i,j}^{n_{1}n_{2}}p_{ij}\;\log_{2}(p_{i}{}_{j})$$
where $p_{ij}$ refers to the occurrence frequency under the relevant events x,y combined together, and $n_{1},n_{2}$ are the possible occurrence states of x,y respectively. In this paper, it refers to the probability of various water shortage situations occurring in the Irrigation Area under the joint impacts of the drought and future water source situation indexes (D) and (S), and H(x,y) refers to the approximate number of early warning signals.

Assuming that $p_{ij}$ is the equal probability given as $p_{ij}$ = $\frac{1}{n_{1}n_{2}}$, then
(4)$$H(x,y)=-\sum_{i,j}^{n_{1}n_{2}}\frac{1}{n_{1}n_{2}}\log_{2}\left(\frac{1}{n_{1}n_{2}}\right)=\log_{2}(n_{1}n_{2}).$$

Assuming that there are $n_{D},n_{S}$ possible states for D and S from the above contents, both $n_{D},n_{S}$ equal to 5 and $H(x,y)=\log_{2}(n_{D}n_{S})=\log_{2}(25)\approx 5$, so five early warning signals are set up, and according to the conventional practice, signals ($m_{0}$) represent green (G, no water shortage), blue (B, mild water shortage), yellow (Y, moderate water shortage), orange (O, serious water shortage), and red lights (R, extreme water shortage), respectively.

##### Calculation of DAI (Drought Alert Index)

The DAI (drought alert index) has combined the drought, future water source situation indexes (D) and (S) in the Irrigation Area, and it reflects a comprehensive index value of the future famine crisis that may arise in the Irrigation Area. It can not only show the drought, but furthermore represents the drought risks possibly existing in the future in combination with the water source situation (the relation between supply and demand). By using Professor Huang Wenzheng’s construction idea on the DAI for reference in this paper, it is thought that the early warning index can be expressed by a nonlinear expression given as $DS^{k}$. The expression of the DAI is
(5)$$DAI=f(D,\;S)\;=\;\log_{n_{D}}\;(D)+k\;\log_{n_{S}}\;(S)$$

In the above formula, $n_{D}=n_{S}=5,D=(1,2,\ldots ,5),$ so 0≤DAI≤k+1,k is an either positive or null integer. DAI shows different signals in different sections, and its upper judgment limit is set as ul:(6)$$ul=k\frac{i-1}{m_{0}-1}+1\;\;(i=1,2,\ldots m_{0};m_{0}=5).$$

When k=2, replace it into the Formula (6), and DAI will be expressed by
(7)$$DAI=\log_{5}\left(DS^{2}\right)\;\;\;D=1,2,\ldots 5;S=1,2,\ldots 5$$

It can be seen that ul=(1,1.5,2,2.5,3) from the calculation through the Formula (7), and then the intervals of early DAI warning index are respectively

0≤DAI≤1; 1≤DAI≤1.5; 1.5≤DAI≤2; 2≤DAI≤2.5 and 2.5≤DAI≤3. Refer to the (Table 3) for the DAI’s alert degrees.

Put different combinations of D and S into the Formula (7) and combine the DAI signals section, then the classification of early warning signals can be obtained, which are represented in the Table 4 as follows:

When *k* > 2, the Formula (6) cannot fully show the early warning signals, so it is determined for *k* = 2 in the Formula (6); hence the Formula (7) stands for the DAI calculation.

##### Time Effect-Integrated Early Drought Warning System

Risks refer to the fact that unexpected events occur in the system in certain circumstances. The risk of the drought alert system established in this paper signifies the event of the inconsistency between the calculations of expected early warning signals and the corresponding inflow probability. If the inflow is sufficient or the water content is rather stable, the early warning signals are easy to determine and are in good compliance with the actual situation generally, but when there are great potential changes in the inflow, for example, the green (G) to red light (R) is obtained when the inflow is from Q_10_ to Q_95_, then it will be difficult to make an eventual early warning decision. Therefore, the risk analysis should be made on the early warning system.

Assuming that the inflow is changed from Q_5_ to Q_95_ in future T_0_ time periods, then the description of early warning index of the different inflow probability is given as shown in Table 5:

From the above Table, the expected DAI can be expressed as
(8)$$E(DAI)=\sum_{t=1}^{n}W_{t}\sum_{\theta_{i}=Q_{5}}^{Q_{95}}p_{t}\;(\theta_{i})\;\log_{5}\;{\left(D_{t}S_{t}^{2}\right)}_{\theta_{i}}.$$

During the drought resistance, affected by the current situation and the satisfaction degree of the future water demand, administrators will pay more attention to the urgent drought degrees at the time when assessment gets closer. It means that during the drought resistance analysis, the drought influence degrees in the first future time (such as the first month) will generally be more significant than that in the second or third month (even in farther occurrence periods). Therefore, focusing on this feature, this paper will assess the time impacts to further explain the early drought warning mechanism.

In the Formula (8), $W_{t}$ is the impact weight ($0\leq W_{t}\leq 1$) of early drought warning at time t, which means the impact degree of the drought analysis at different time scales. Focusing on the weight calibration analysis, this paper adopts the planed hydrological Recession Curve concept to define the parameter λ of the planed Recession Constant.

First, assume that
(9)$$\frac{\mathrm{df}}{\mathrm{dt}}=-\lambda f$$
in which, f is given by
(10)Lnf=−λt+constant.

If $f=f_{0}$ (t = 1), the following expression is obtained:(11)$$f=f_{0}e^{-\lambda (t-1)}.$$

Replacing it into the Formula (8), we find
(12)$$DAI=\frac{1}{\sum_{t=1}^{n}e^{-\lambda (t-1)}}\sum_{t=1}^{n}\left[\log_{5}{\left(D_{t}S_{t}^{2}\right)}_{\theta_{i}}\right]e^{-\lambda (t-1)}.$$

However, the impact weight $W_{t}$ at time t can be expressed as follows:(13)$$W_{t}=\frac{e^{-\lambda (t-1)}}{\sum_{t=1}^{n}e^{-\lambda (t-1)}}.$$

With the increase and decrease of λ, the $W_{t}$ will generate different changes in impacts. As it is shown in the Table 2, when *n* = 3, λ will progressively increase, and the weight will gradually increase when *t* = 1; contrariwise, the weight will progressively decrease when *t* = 3. This characteristic can just explain the urgency degree at different times of analysis during the drought resistance: “The urgency degree is not only affected by the satisfaction degree of the water demand, but also the degree of its influences on the crisis should be intensified as the analysis point of the drought resistance time gets closer (see Figure 2 and Figure 3).

For the consideration of three future time periods (*n* = 3) in this paper, through the setting of different intervals of λ = 0.0~2.0 and $W_{t}$, and the analysis and discussion of the past abundant and dry hydrological events, the parameter λ is set as 0.2, so the weights are $W_{t}$ = 0.40, $W_{t}$ = 0.33, and $W_{t}$ = 0.27, respectively.

According to the calculation of the above indexes, the decision-making system for early drought warning can be set up in the future time periods. By considering the time interval impacts, when the cumulative frequency at time t in the future is P and the future water source situation is $\theta_{i}$, the DAI containing the time effect can be obtained as shown in the following:(14)$$DAI_{P}=\frac{e^{-\lambda (t-1)}}{\sum_{t=1}^{n}e^{-\lambda (t-1)}}\sum_{t=1}^{n}\left[\log_{5}{\left(D_{t}S_{t}^{2}\right)}_{\theta_{i}}\right]$$

The DAI consists of the drought and water source indexes, (D) and (S), and is constructed through the information entropy theory. Based on the statistical method, we can determine the early warning signals and alert division and construct the drought alert mechanism in combination with the time effect integration.

## 3. Case Studies on Early Drought Warning System in Irrigation Areas

Based on the collected information about the precipitation, soil moisture, water diversion amount from channel head, and amount of groundwater mining, the comprehensive evaluation on the drought occurrence and the drought degree of each month in 2013 was carried out; in combination with the water source situation from the abundant to dry situation possibly occurring, the analogue simulation of early drought warning was also carried out, and the comparison between the simulation results and the results monitored by the Jinghui Channel Management Office was achieved.

### 3.1. Calculation of Single Factor Index

#### 3.1.1. Precipitation Index

Use the SPI calculation method to obtain the estimated SPI values of each month. Since the SPI indexes have a lot of limits to the sequences length, the data of only one year cannot bring rather accurate results. Therefore, this paper cites the precipitation in the Irrigation Area from 1961 to 2013, from which the evaluation was carried out and the adaptation to the accuracy requirements for the SPI evaluation was done. After the evaluation, the drought degree of each month in 2013 was obtained, as illustrated in Table 6.

#### 3.1.2. Soil Moisture

The soil moisture data was extracted from the data provided by the Administration Office in the Jinghui Channel Irrigation Area. This Bureau offers the soil moisture of the Louzi Village of Zhongzhang Town, test station, and Gedong Village of Xinshi Town. During the data selection process, the soil mass water content before winter wheat and summer corn jointing were selected as the objects which were measured at the depth of 40 cm below the surface, and the mean values after jointing were taken as the reference at 1 m spot. Based on the soil moisture analysis situation in the Irrigation Area for the past years, the suitable evaluation standard was determined. Table 7 shows the monthly soil moisture situation in 2013.

### 3.2. Membership Function Determination and Fuzzy Matrix Establishment

Use the membership function establishment method to set up the precipitation and soil moisture membership functions. Refer to the Table 8 for the determination of the corresponding membership function levels.

#### 3.2.1. Membership Function of the Precipitation

(15)$$U_{V_{1}}(x)=\left\{\begin{array}{c} 1\;\;(x\geq -0.5)\\ \frac{x+0.75}{-0.5+0.75}(-0.75<x<-0.5)\\ 0\;\;(x\leq -0.75) \end{array}\right.$$

(16)$$U_{V_{2}}(x)=\left\{\begin{array}{c} \frac{-0.5-x}{-0.5+0.75}\;(-0.75\leq x<-0.5)\\ \frac{x-k_{3}}{-0.75-k_{3}}\;(-1.25<x<-0.75)\\ 0\;\;(x\leq -1.25,x\geq -0.5) \end{array}\right.$$

(17)$$U_{V_{3}}(x)=\left\{\begin{array}{c} \frac{-0.75-x}{-0.75+1.25}\;(-1.25\leq x<-0.75)\\ \frac{x+1.75}{-1.25+1.75}\;(-1.75<x<-1.25)\\ 0\;\;\;(x\leq -1.75,x\geq -0.75) \end{array}\right.$$

(18)$$U_{V_{4}}(x)=\left\{\begin{array}{c} \frac{-1.25-x}{-1.25+1.75}\;(-1.75\leq x<-1.25)\\ \frac{x+2}{-1.75-k_{4}}\;\;(-2<x<-1.75)\\ 0\;\;\;(x\leq -2,x\geq -1.75) \end{array}\right.$$

(19)$$U_{V_{5}}(x)=\left\{\begin{array}{c} 0\;\;\;(x\geq -1.75)\\ \frac{-1.75-x}{-1.75+2}(-2<x<-1.75)\\ 1\;\;\;\;(x\leq -2) \end{array}\right.$$

#### 3.2.2. Membership Function of the Soil Moisture

(20)$$U_{V_{1}}(x)=\left\{\begin{array}{c} 1\;\;\;(x\geq 22)\\ \frac{x-21.25}{22-21.25}\;(21.25<x<22)\\ 0\;\;(x\leq 21.25) \end{array}\right.$$

(21)$$U_{V_{2}}(x)=\left\{\begin{array}{c} \frac{22-x}{22-21.25}\;\;(21.25\leq x<22)\\ \frac{x-19.75}{21.25-19.75}\;(19.75<x<21.25)\\ 0\;\;\;(x\leq 19.75,x\geq 22) \end{array}\right.$$

(22)$$U_{V_{3}}(x)=\left\{\begin{array}{c} \frac{21.25-x}{21.25-19.75}\;(19.75\leq x<21.25)\\ \frac{x-17.75}{19.75-17.75}\;(17.75<x<19.75)\\ 0\;\;(x\leq 17.75,x\geq 21.25) \end{array}\right.$$

(23)$$U_{V_{4}}(x)=\left\{\begin{array}{c} \frac{19.75-x}{19.75-17.75}\;(17.75\leq x<19.75)\\ \frac{x-16.5}{17.75-16.5}\;(16.5<x<17.75)\\ 0\;\;\;(x\leq 16.5,x\geq 17.75) \end{array}\right.$$

(24)$$U_{V_{5}}(x)=\left\{\begin{array}{c} 0\;\;\;(x\geq 17.75)\\ \frac{17.75-x}{17.75-16.5}(16.5<x<17.75)\\ 1\;\;\;(x\leq 16.5) \end{array}\right.$$

#### 3.2.3. Fuzzy Matrix

Take the Irrigation Area in January 2013 for an example to set up the fuzzy matrix as follows:(25)$$R_{1}=\left[\begin{array}{ccccc} r_{11} & r_{12} & r_{13} & r_{14} & r_{15}\\ r_{21} & r_{22} & r_{23} & r_{24} & r_{25} \end{array}\right]=\left[\begin{array}{ccccc} 1.000 & 0.000 & 0.000 & 0.000 & 0.000\\ 0.780 & 0.220 & 0.000 & 0.000 & 0.000 \end{array}\right]$$

The fuzzy matrix $R_{2}-R_{12}$ from February to December in 2013 can be similarly set up.

### 3.3. Determination of Fuzzy Weight Vector

For the Jinghui Channel Irrigation Area, the soil moisture is the most direct embodiment of the drought and also a key factor influencing the crops growth. Precipitation is the fundamental factor which causes the drought in the Irrigation Area; however, it does not mean that the drought will be definitely caused when the precipitation is less, which has a lot to do with the water demand for crops. Therefore, taking all above factors into consideration, the weight distribution value of the precipitation and soil moisture in the Jinghui Channel Irrigation Area is given by: $A=(a_{1},a_{2})$ = (0.45, 0.55).

### 3.4. Drought Evaluation Index (D)

Taking the Jinghui Channel in January 2013 for an example to establish the fuzzy synthetic evaluation model, the result was obtained as
(26)$$AR=(0.45,0.55)\left[\begin{array}{ccccc} 1.000 & 0.000 & 0.000 & 0.000 & 0.000\\ 0.780 & 0.220 & 0.000 & 0.000 & 0.000 \end{array}\right]\;=(0.878,\;0.122,\;0.000,\;0.000,\;0.000)=B$$

According to the maximum membership principle, the drought evaluation result in the Irrigation Area in January 2013 is no drought, and the corresponding membership degree is 0.878.

Similarly, the results from February to December were also obtained. Refer to the Table 9, Table 10 and Table 11 for the specific results information.

By replacing the data in Table 9 and Table 10 into the fuzzy synthetic evaluation model, the drought levels from January to December in 2013 were obtained, as shown in Table 11:

### 3.5. Future Water Source Situation Index (S)

Based on the analysis of the monthly channel head’s water diversion amount from 1994 to 2013 and the corresponding 2013 irrigation system, the monthly water source situation indexes in 2013 were set up. Specific steps are the following:(1)Execute an empirical frequency analysis in order to determine the monthly channel head’s water diversion amount from 1994 to 2013.(2)Determine the amount of monthly groundwater mining in the Irrigation Area based on the amount available in spring, summer, and winter irrigations during the water demand, and use analysis in the Irrigation Area from 1994 to 2013.(3)Add the monthly channel head’s water diversion and groundwater mining amounts to get the inflow in the Irrigation Area.(4)Check the monthly water demand in corresponding years according to the irrigation systems in different hydrological years in the Irrigation Area. The year 2013 studied in this paper is a moderate-drought year, so the corresponding irrigation system and water demand when P is equal to 75% was selected.(5)The difference value between the monthly water demand and possible inflow under different frequencies is the water deficit, and the water deficit/water demand is the water deficiency ratio. The water source situation index level (S) can be found in the (Table 12) as it is designed to represent it as follows:

### 3.6. Early Drought Warning in 2013

#### 3.6.1. Early Drought Warning

The data of drought situation index (D) and future water source situation index (S) of the irrigation area in 2013 were collected to calculate the early drought warning under various water source situations. Then, the corresponding early warning signals were obtained, which are illustrated in Table 13.

#### 3.6.2. Results Analysis

A comparison of the early drought warning simulation results with the Jinghui Channel Irrigation Area’s monitoring ones released by officials is shown in Table 14.

From the Table 13, the early warning system results constructed in this article for droughts in the Jinghui Channel Irrigation Area are basically consistent with the monitoring information released by Jinghui Channel Administration Office, not only embodying the drought situation at each stage, but also correctly describing the drought’s potential development trend in three future months so that administrators can work the corresponding measures out.

September and October experience the harvest stage for the summer corn and sowing stage for the winter wheat, respectively, and abundant water exerts no impacts on crops. So the early warning signal shows the “green light”, but this kind of situation will have some impacts on the growth of winter wheat at the next stage and the water moisture storage to some extent. Therefore, the drought indexes (D) are suggested to be considered for early drought warning in September and October to provide administrators with more comprehensive and practical information about it.

Table 15 illustrates the water supply analysis in the Irrigation Area for the three future months from March 2013, and Table 16 stands for the same analysis from September 2013.

$DAI_{20\%}$ = 0.40 × 0.86 + 0.33 × 1 + 0.27 × 0.43 = 0.790 (the early warning signal shows the green light under this frequency.)

$DAI_{50\%}$ = 0.40 × 1.72 + 0.33 × 3.00 + 0.27 × 1.80 = 2.164 (the signal shows the orange light under this frequency.)

$DAI_{80\%}$ = 0.40 × 2.00 + 0.33 × 3.00 + 0.27 × 2.15 = 2.371 (It shows the orange light again.)

$DAI_{95\%}$ = 0.40 × 2.00 + 0.33 × 3.00 + 0.27 × 2.43 = 2.446 (orange light is also shown by the signal under this frequency.)

$DAI_{20\%}$ = 0.40 × 0.86 + 0.33 × 0 + 0.27 × 0.86 = 0.576 (the early warning signal shows the green light under this frequency.)

$DAI_{50\%}$ = 0.40 × 0.86 + 0.33 × 0 + 0.27 × 0.86 = 0.576 (the signal also shows the green light.)

$DAI_{80\%}$ = 0.40 × 0.86 + 0.33 × 0 + 0.27 × 1.36 = 0.711 (It emits the green light for this frequency also.)

$DAI_{95\%}$ = 0.40 × 0.86 + 0.33 × 0 + 0.27 × 1.72 = 0.808 (For this frequency, the green light is shown too.)

#### 3.6.3. Comparison of Results

(1)Meteorological drought grades and actual early warning signals comparison

The Figure 2, Figure 3, Figure 4 and Figure 5 stand for the green, blue, yellow, orange, and red lights, respectively. From the diagram, the actual early warning signals are not consistent with precipitation-related drought degrees, and for the Jinghui Channel Irrigation Area, the precipitation amount explains a fundamental reason for the drought, but it is not the only decisive factor. Only when the relationship between the soil moisture and the balance between supply and demand is also considered can the realistic information about early drought warning be obtained.

To be specific, although the precipitation in February and March is less compared with previous years, the water source situation is quite ideal, so the actual early warning signal is the “green light”. The precipitation from April to July is comparatively abundant, but because the winter wheat experiences the advantage, germination, flowering, soft dough, and seven-leaf stages, and is just at the stage of needs for much growth water, so the actual early warning signal shows the early waning state and even sends the highest early warning the “red light” in May, which should arouse the administrators’ attention. In August, precipitation is less and the meteorological index shows the extreme drought, but since the water source situation analysis has been carried out, the actual early warning signal shows the “yellow light”. From September to December, meteorological drought grades are basically consistent with the actual early drought warning.

The above-mentioned comparisons indicate that the precipitation amount is one of the factors influencing the early drought warning, but the warning system constructed in this article is more comprehensive.

(2)Soil moisture-related drought degrees and actual early warning signals comparison

From the diagram, there are similarities between the soil moisture-related drought degrees within the year and the actual early warning signal variation trends, but they are not consistent. For the Jinghui Channel Irrigation Area, the soil moisture is the most direct manifestation of the drought degrees, so its variation trend will definitely influence that of the early drought warning, but both the precipitation and the relationship between the supply and demand also impact the early drought warning results.

To be specific, the soil moisture in January and February is quite ideal, and the actual early drought warning shows the “green light”; the soil water deficit in March is quite severe, but the water source situation is ideal, so the actual early warning shows the “green light”, and from April to August, the actual early warning signals are basically consistent with the soil moisture’s drought degrees. After September, all actual early warning signals show “green lights”, but the soil moisture is not quite ideal, and this is because September and October are the harvest time for summer corn and the sowing time for winter wheat, respectively, and the water moisture shortage exerts no impact on the harvest. Also, winter wheat is at the growth stage in November and December and does not have a high water demand, so even though the soil moisture is comparatively poor, early drought warning results are quite optimistic.

In general, soil moisture is the most direct but not the only decisive factor affecting the drought in the Irrigation Area and has a greater impact on the early warning.

(3)Drought grades (D) and actual early warning signals comparison

From the analysis of the figure, the drought index (D) takes place after the precipitation and soil moisture are comprehensively considered, and a certain correlation exists between the early drought warning and D. The drought in January, February, May to July, November, and December shows quite consistently, but remains different in the other months because the channel head’s water diversion amount and groundwater mining and other incoming cases are considered during the early drought warning, also personifying the human activities’ regulating effect on the early drought warning. The above conclusions are drawn from the differences between the water source situation indexes and drought grades variations (see Figure 6).

From all comparison results, the early drought warning is a comprehensive system, liable to change when it is affected by its component elements, but owing to the comprehensive embodiment of natural factors and human activities, the results are more comprehensive, objective, and practical.

## 4. Conclusions

A basic framework of the early drought warning system is constructed in this article. On the basis of basic theories analysis regarding home and abroad early drought warning, the early drought warning definition and meaning are elaborated, and evaluations, forecasts, and early warning are also pointed out. The early drought warning index which consists of the drought and water source situation indexes, (D) and (S), and which is based on the information entropy theory is put forward. The grades of the early agricultural drought disaster warning which show “the green, blue, yellow, orange, and red lights” are set up. Also, according to the statistical approach, the early warning signal, warning zone plotting, and early drought warning system integrating the time effect are determined.

Conclusions are the following:(1)The early drought warning’s simulated values and results are basically consistent with the monitoring values and information released by the Jinghui Channel Administration Office for the Irrigation Area, not only embodying the drought situation at each stage, but also correctly describing the drought’s potential development trend in three future months.(2)The actual early warning signals are not consistent with the precipitation-related drought degrees, and for the Jinghui Channel Irrigation Area, the precipitation amount is considered to be a fundamental reason for the drought in the Area, but it is not the only decisive factor since the only relationship between the soil moisture and the balance between supply and demand also has been considered, so the realistic information about early drought warning can be obtained.(3)There are similarities between the soil moisture–related drought degrees within the year and the actual early warning signals variation trends, but they are not consistent. For the Jinghui Channel Irrigation Area, the soil moisture is the most direct manifestation of the drought degree, so the soil moisture’s variation trend will definitely influence that of the early drought warning, but both the precipitation and the relationship between the supply and demand will also have an impact on the early drought warning results.(4)Soil moisture is the most direct factor impacting on the drought situation in irrigation districts, and it has a relatively larger impact on the drought warning, but it is not the exclusive determining factor. Drought indicator (D) has taken into account the amount of precipitation and soil moisture. The drought warning has a certain correlation with D. The drought performance in this study was consistent in January, February, May to July, November, and December, whereas there were some differences in the rest of the year. This is because in the calculation process of the drought warning, incoming water conditions such as the headwater diversion and the groundwater extraction were considered, which reflected the regulating role of human activities in the process of drought warning.

## Figures and Tables

**Figure 1 ijerph-17-00374-f001:**
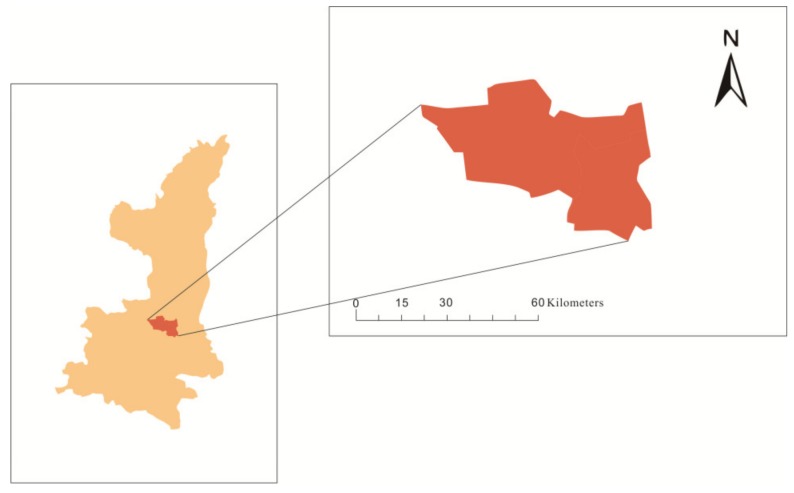
The location of Jinghui irrigation area in Shannxi Province.

**Figure 2 ijerph-17-00374-f002:**
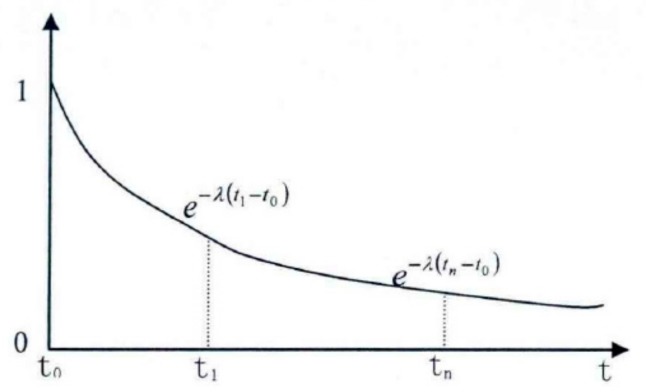
Diminishing curve reflecting the future time influence.

**Figure 3 ijerph-17-00374-f003:**
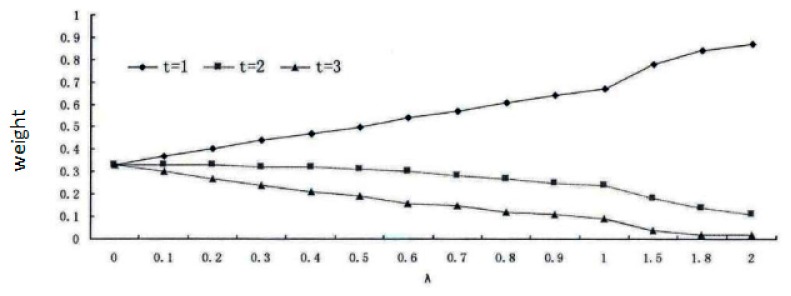
Variations of weight with the parameter λ.

**Figure 4 ijerph-17-00374-f004:**
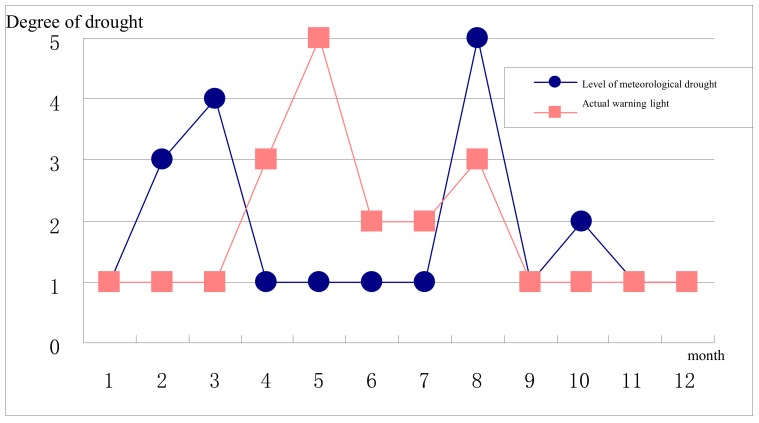
Monthly precipitation-related drought degrees and actual early warning signals in 2013.

**Figure 5 ijerph-17-00374-f005:**
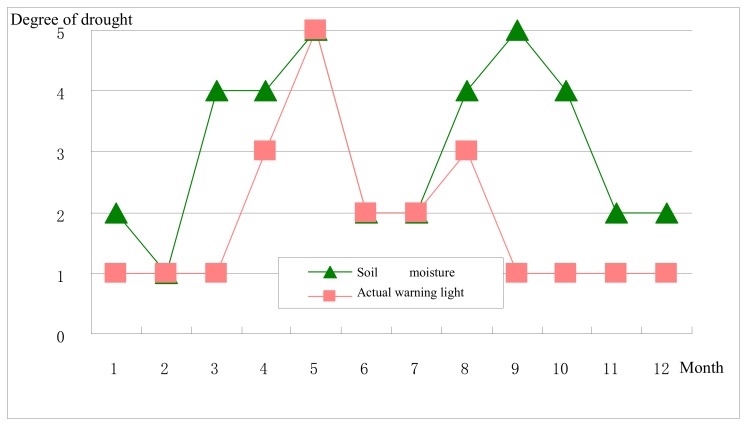
Monthly soil moisture drought degrees and actual early warning signals in 2013.

**Figure 6 ijerph-17-00374-f006:**
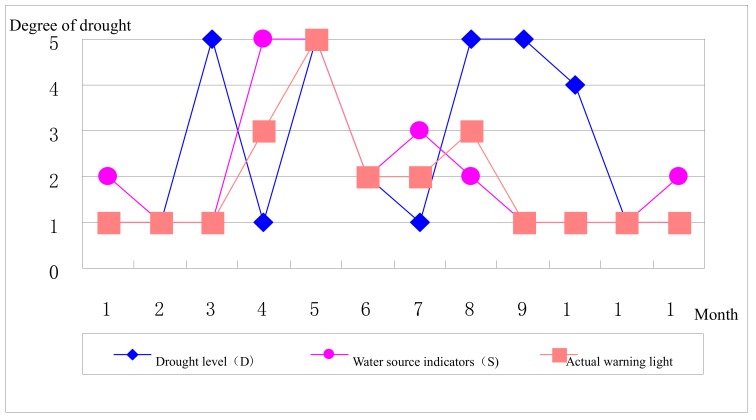
Monthly drought grades and actual early warning signals in 2013.

**Table 1 ijerph-17-00374-t001:** Classification of drought grades according to evaluation indexes.

Code C	Index	Unit	Evaluation Standard				
			No Drought	Mild Drought	Moderate Drought	Serious Drought	Extreme Drought
C1	SPI	Dimensionless	>−0.5	−1.0~−0.5	−1.5~−1.0	−2.0~−1.5	≤−2.0
C2	Soil moisture	%	≥22.0	20.5~22.0	19.0~20.5	16.5~19.0	<16.5

**Table 2 ijerph-17-00374-t002:** Water source situation index grades.

Water Shortage Grade		Water Shortage Rate = Water Shortage/Water Demand × 100%	
		Agricultural Water	Water for Public Use (Multi-Purpose Reservoir, Including Agriculture)
Water shortage degree in the future	No water shortage	0	0
	Mild water shortage	0~30%	0~10%
	Moderate water shortage	30%~40%	10%~20%
	Serious water shortage	40%~50%	20%~30%
	Extreme water shortage	>50%	>30%

**Table 3 ijerph-17-00374-t003:** Determination of DAI (drought alert index) alert intervals.

Early Warning Signals	Green Light (G)	Blue Light (B)	Yellow Light (Y)	Orange Light (O)	Red Light (R)
Early warning index intervals	0≤DAI≤1	1≤DAI≤1.5	1.5≤DAI≤2	2≤DAI≤2.5	2.5≤DAI≤3
Alert degrees	Normal state	Alert	Raising alert	High alert	Severe alert

**Table 4 ijerph-17-00374-t004:** Values of DAI and classification of early warning signals.

Drought Index (D)	Future Water Source Situation Index (S)				
	1 (No Water Shortage)	2 (Mild Water Shortage)	3 (Moderate Water Shortage)	4 (Serious Water Shortage)	5 (Extreme Water Shortage)
1. No drought	0 (G)	0.86 (G)	1.36 (B)	1.72 (Y)	2.00 (Y)
2. Mild drought	0.43 (G)	1.29 (B)	1.80 (Y)	2.15 (O)	2.43 (O)
3. Moderate drought	0.68 (G)	1.54 (Y)	2.05 (O)	2.41 (O)	2.68 (R)
4. Serious drought	0.86 (G)	1.72 (Y)	2.23 (O)	2.58 (R)	2.86 (R)
5. Extreme drought	1 (G)	1.86 (Y)	2.37 (O)	2.72 (R)	3.00 (R)

**Table 5 ijerph-17-00374-t005:** Analysis of early drought warning for different inflow probability time periods.

Different Inflow Situation $\theta_{i}$	Different Inflow Situations Probability $p_{t}(\theta_{i})$	DAI (t = 1)	DAI (t = 2)	…	DAI (t = n)
$Q_{5}$	$p_{t}(Q_{5})$	$\log_{5}{\left(D_{1}S_{1}^{2}\right)}_{Q5}$	$\log_{5}{\left(D_{2}S_{2}^{2}\right)}_{Q5}$	…	$\log_{5}{\left(D_{n}S_{n}^{2}\right)}_{Q5}$
$Q_{10}$	$p_{t}(Q_{10})$	$\log_{5}{\left(D_{1}S_{1}^{2}\right)}_{Q10}$	$\log_{5}{\left(D_{2}S_{2}^{2}\right)}_{Q10}$	…	$\log_{5}{\left(D_{n}S_{n}^{2}\right)}_{Q10}$
$Q_{20}$	$p_{t}(Q_{20})$	$\log_{5}{\left(D_{1}S_{1}^{2}\right)}_{Q20}$	$\log_{5}{\left(D_{2}S_{2}^{2}\right)}_{Q20}$	…	$\log_{5}{\left(D_{n}S_{n}^{2}\right)}_{Q20}$
…	…	…	…	…	…
$Q_{95}$	$p_{t}(Q_{95})$	$\log_{5}{\left(D_{1}S_{1}^{2}\right)}_{Q95}$	$\log_{5}{\left(D_{2}S_{2}^{2}\right)}_{Q95}$	…	$\log_{5}{\left(D_{n}S_{n}^{2}\right)}_{Q95}$

**Table 6 ijerph-17-00374-t006:** Precipitation and SPI index values of each month in 2013 in Jinghui Channel Irrigation Area.

Month	Precipitation (mm)	SPI Value
1	5.0	−0.0528
2	6.7	−1.0828
3	0	−1.9808
4	38.8	−0.1303
5	50.1	0.2975
6	78.6	0.0687
7	100.5	0.7109
8	6.4	−2.2655
9	36.6	−0.2556
10	6.9	−0.9234
11	4.6	0.4583
12	18.5	1.9725
Total	352.7	

**Table 7 ijerph-17-00374-t007:** Soil Moisture of Each Month in 2013.

Month	1	2	3	4	5	6	7	8	9	10	11	12
Soil moisture %	21.83	22.83	18.50	18.90	16.40	21.20	20.73	18.23	16.44	18.18	21.62	21.40

**Table 8 ijerph-17-00374-t008:** Determination of Membership Function Levels corresponding to Indexes.

Index	Index System	Unit	Determination of the Membership Function Levels Defined through the Triangular Fuzzy Distribution Method				
			K_1_	K_2_	K_3_	K_4_	K_5_
	SPI	No dimension	−0.5	−0.75	−1.25	−1.75	−2.0
	Soil moisture	%	22	21.25	19.75	17.75	16.5

**Table 9 ijerph-17-00374-t009:** Monthly precipitation membership degree for the Jinghui Channel Irrigation Area in 2013.

Month	SPI	Drought Levels				
		No Drought	Mild Drought	Moderate Drought	Serious Drought	Extreme Drought
1	−0.0528	1.000	0.000	0.000	0.000	0.000
2	−1.0828	0.000	0.334	0.666	0.000	0.000
3	−1.9808	0.000	0.000	0.000	0.077	0.923
4	−0.1303	1.000	0.000	0.000	0.000	0.000
5	0.2975	1.000	0.000	0.000	0.000	0.000
6	0.0687	1.000	0.000	0.000	0.000	0.000
7	0.7109	1.000	0.000	0.000	0.000	0.000
8	−2.2655	0.000	0.000	0.000	0.000	1.000
9	−0.2556	1.000	0.000	0.000	0.000	0.000
10	−0.9234	0.000	0.653	0.347	0.000	0.000
11	0.4583	1.000	0.000	0.000	0.000	0.000
12	1.9725	1.000	0.000	0.000	0.000	0.000

**Table 10 ijerph-17-00374-t010:** Monthly soil moisture membership degree for the Jinghui Channel Irrigation Area in 2013.

Month	Soil Moisture	Drought Levels				
		No Drought	Mild Drought	Moderate Drought	Serious Drought	Extreme Drought
1	21.83	0.780	0.220	0.000	0.000	0.000
2	22.83	1.000	0.000	0.000	0.000	0.000
3	18.50	0.000	0.000	0.375	0.625	0.000
4	18.90	0.000	0.000	0.575	0.425	0.000
5	16.40	0.000	0.000	0.000	0.000	1.000
6	21.20	0.000	0.967	0.033	0.000	0.000
7	20.73	0.000	0.656	0.344	0.000	0.000
8	18.23	0.000	0.000	0.242	0.758	0.000
9	16.44	0.000	0.000	0.000	0.000	1.000
10	18.18	0.000	0.000	0.217	0.783	0.000
11	21.62	0.489	0.511	0.000	0.000	0.000
12	21.40	0.200	0.800	0.000	0.000	0.000

**Table 11 ijerph-17-00374-t011:** Monthly drought evaluation results for the Jinghui Channel Irrigation Area in 2013.

Month	Drought Levels					Maximum Membership Degree	Drought Evaluation
	No Drought 1	Mild Drought 2	Moderate Drought 3	Serious Drought 4	Extreme Drought 5		
1	0.878	0.122	0.000	0.000	0.000	0.878	1
2	0.550	0.150	0.300	0.000	0.000	0.550	1
3	0.000	0.000	0.206	0.378	0.415	0.415	5
4	0.450	0.000	0.316	0.234	0.000	0.450	1
5	0.450	0.000	0.000	0.000	0.550	0.550	5
6	0.450	0.532	0.018	0.000	0.000	0.532	2
7	0.450	0.361	0.189	0.000	0.000	0.450	1
8	0.000	0.000	0.133	0.417	0.450	0.450	5
9	0.450	0.000	0.000	0.000	0.550	0.550	5
10	0.000	0.294	0.276	0.431	0.000	0.431	4
11	0.719	0.281	0.000	0.000	0.000	0.719	1
12	0.560	0.440	0.000	0.000	0.000	0.560	1

**Table 12 ijerph-17-00374-t012:** Water Source Situation in the Irrigation Area.

Water Shortage Level		Water for Agricultural Use
S_i_ Future water shortage degree	No water shortage	0
	Mild water shortage	0–30%
	Moderate water shortage	30–40%
	Serious water shortage	40–50%
	Extreme water shortage	>50%

**Table 13 ijerph-17-00374-t013:** Early Drought Warning Results for the Irrigation Area in 2013.

Month			1	2	3	4	5	6	7	8	9	10	11	12
Drought Index Levels			1	1	5	1	5	2	1	5	5	4	1	1
Water source situation levels	Q_5_	Sufficient state Dry state	1	1	1	2	1	1	2	2	1	1	1	2
	Q_10_		1	1	1	2	2	2	2	2	1	1	1	2
	Q_20_		1	1	1	2	2	2	2	2	1	1	1	2
	Q_30_		1	1	1	2	2	2	2	2	1	1	1	2
	Q_40_		1	1	1	4	5	3	3	2	1	1	1	2
	Q_50_		1	1	2	4	5	3	3	3	1	1	1	2
	Q_60_		2	1	2	5	5	3	3	3	1	1	1	3
	Q_70_		2	1	2	5	5	3	4	3	1	1	1	3
	Q_80_		2	1	2	5	5	4	4	4	1	1	1	3
	Q_90_		2	1	3	5	5	4	5	4	1	1	1	4
	Q_95_		2	1	5	5	5	5	5	4	1	1	1	4
	Actual inflow		2	1	1	5	5	2	3	2	1	1	1	2
Early warning signals for drought	Q_5_	Sufficient state Dry state	1-G	1-G	1-G	1-G	1-G	1-G	1-G	3-Y	1-G	1-G	1-G	1-G
	Q_10_		1-G	1-G	1-G	1-G	3-Y	2-B	1-G	3-Y	1-G	1-G	1-G	1-G
	Q_20_		1-G	1-G	1-G	1-G	3-Y	2-B	1-G	3-Y	1-G	1-G	1-G	1-G
	Q_30_		1-G	1-G	1-G	1-G	3-Y	2-B	1-G	3-Y	1-G	1-G	1-G	1-G
	Q_40_		1-G	1-G	1-G	3-Y	5-R	3-Y	2-B	4-O	1-G	1-G	1-G	1-G
	Q_50_		1-G	1-G	3-Y	3-Y	5-R	3-Y	2-B	4-O	1-G	1-G	1-G	2-B
	Q_60_		1-G	1-G	3-Y	3-Y	5-R	3-Y	2-B	4-O	1-G	1-G	1-G	2-B
	Q_70_		1-G	1-G	3-Y	3-Y	5-R	3-Y	3-Y	4-O	1-G	1-G	1-G	2-B
	Q_80_		1-G	1-G	3-Y	3-Y	5-R	4-O	3-Y	5-R	1-G	1-G	1-G	2-B
	Q_90_		1-G	1-G	4-O	3-Y	5-R	4-O	3-Y	5-R	1-G	1-G	1-G	3-Y
	Q_95_		1-G	1-G	5-R	3-Y	5-R	4-O	3-Y	5-R	1-G	1-G	1-G	3-Y
	Actual levels		1	1	1	3	5	2	2	3	1	1	1	1
	Actual signals		G	G	G	Y	R	B	B	Y	G	G	G	G

**Table 14 ijerph-17-00374-t014:** Comparison between 2013 early drought warning and Jinghui Channel Irrigation Area’s monitoring results.

Month	Early Drought Warning Analysis	Official Monitoring Results Released by the Jinghui Channel Management Bureau	Remark
January	Water supply under all frequencies in the Irrigation Area is optimistic, so all signals analysis results shows the “green light”, and the potential drought development crisis is improvable.	The winter irrigation soil moisture is sufficient with a small evaporation amount, fully satisfying the wheat requirements for living through the winter.	The soil moisture in this month is adequate, but the temperature is low. Considering the freezing elimination effect’s impacts, a moderate winter wheat’s water filing amount irrigated based on stubble repeating is suggested.
February	It is with the same case as January; the drought severity and water supply in this month are optimistic, and all signals show the “green lights”.	Winter irrigation has been carried out in the Irrigation Area, and the soil moisture is sufficient and with a small evaporation amount, which is quite favorable for the wheat to live through the winter.	Timely provision of fertilizers and water to the wheat for which winter irrigation has not been carried out.
March	When Q < Q_40_, the signal shows the “yellow light”, representing the three future months’ drought development may cause a water supply scant, so further analysis is carried out during that time. Table 10 analyzes the three future months water supply (April, May, and June) and finds that the signal shows the “orange light” when Q (the estimated water source situation) is strictly less than Q_40_, meaning that it has been necessary for the administration units to make big moves to provide measures regarding agricultural water supply, otherwise unfavorable impacts on the crop growth will be caused for the in-time water supply shortage.	The soil moisture for which winter irrigation has been carried out in the Irrigation Area satisfies the requirements for the winter wheat growth, but it quickly declines owing to the wheat growth and the rapid temperature rise; if there is no recent precipitation, unfavorable impacts on the winter wheat jointing will be caused on the basis of current declining situation.	The drought in this month is severe, but the water source situation analysis is carried out and the early drought warning index is inconsistent with the drought index, which embodies that the regulating effects of the human activities in early drought warning exist and exert a critical impact in the Irrigation Area.
April	The drought is in a “drought-free” state, but when Q < Q_30_, the signal begins to show the “yellow light” and represent the water source situation in this month cannot well satisfy the crops’ water demands, so administrators need to take timely measures concerning the agricultural water supply to guarantee the normal crops growth.	Winter wheat enters the advantage and germination stages, respectively, sensitive to the moisture, needing quite a large amount of water, and the soil moisture is greatly declining. If there is no evident precipitation later, it will be quite unfavorable to wheat filling, thus affecting the output harvest and winter wheat quality.	With the constant temperature rise in this month and the increasing evaporation amount, and the water demands for winter wheat during the growth stage, administrators should timely feed water.
May	The drought’s development trend has been clear: The drought is very severe; when Q (the future water source situation) is strictly less than Q_10_, the signal begins to show the “yellow light”, and when Q (the estimated water source situation) is inferior to Q_40_, the early drought warning signal shows the “red light”, so administrators need to urgently take large-scale measures regarding the agricultural water supply, otherwise it will be extremely unfavorable to the crops growth in the Irrigation Area.	Winter wheat enters the flowering period, so it has a great water demands; the soil moisture range is very large, and although 15.5 mm precipitation is provided, the requirements for normal growth are hard to be satisfied, which is unfavorable to the wheat filling.	This is a quite high water demand month for the winter wheat, so administrators should well prepare measures regarding the water supply in advance.
June	The drought’s development trend is slightly weakened compared with that in May; when (the future water source situation) Q < Q_80_, the signal begins to show the “orange light”, which may have something to do with the drought in June. Since the drought grade is “mild”, it can be seen that there will be comparatively abundant precipitation, and the imbalance between supply and demand in the Irrigation Area is relatively weakened, and the early drought warning signals will reduce accordingly.	Winter wheat has entered the soft dough stage, so the soil moisture impact on the harvest will decrease in quantity, and also the precipitation in this month is relatively abundant. Although the drought has certain impacts on the winter wheat harvest and the summer corn sowing, it is favorable to the summer corn germination.	The winter wheat does not have a high water demand in this month, and the summer corn enters the sowing and emergence stage.
July	The drought trend declines to some degree, but for semiarid areas in north, it is just in the period when the temperature is high and the evaporation amount is large, so even though the water source situation is sufficient, the loss among it should be considered. Therefore, an attention should be paid to the early drought warning result when the water source situation is less. When Q (the future water source situation) is less than Q_70_, the signal shows the “yellow light” and administrators should pay attention to it too.	The summer corn experiences the seven-leaf and jointing stages, respectively, in this month, which is also the period when the water demand becomes the highest, but the precipitation is up to 100.5 mm. Summer corn grows quite well, but in paddy field pieces without irrigation, the moisture content is quite low and timely water supply is necessary.	In this month, the temperature is high and the evaporation amount is large, and also it is the growth stage, when the summer corn has a quite high water demand.
August	The drought’s development trend goes up to some extent, and the drought index is in the “extreme drought” state, which may be caused by less precipitation and low moisture content in the soil this month together with the large moisture evaporation amount in summer, so the drought is intensified, and administrators are suggested to timely refill water in the Irrigation Area.	Summer corn experiences the flowering and filling stages, respectively, and has a quite high water demand. If the soil water content is insufficient, inadequate filling and imperfect grains will be caused, thus giving rise to the harvest reduction. Owing to water storage in crops and huge field transpiration in the early period, the soil water content greatly declines, so water supplement is suggested.	This month is the period in which water demands are the highest for the summer corn and it is also a key stage to decide whether the harvest is good or not.
September	The drought crisis has completely disappeared, and the early drought warning is a clear “green light”; hence the drought is “extreme”, but the water source situation is extremely optimistic, indicating that the supply and demand is currently quite optimistic and water sources can relieve the drought impacts, and these are the drought trends which need to be further explored. The three future months’ water supply analysis in the Irrigation Area is carried out to determine the potential drought crises. Table 1 tells that the expected in-time early warning indexes are quite ideal, and from Q > Q_30_ to Q > Q_80_, signals are all green lights, which is not quite related to the crops’ water demand in this stage.	Summer corn in the Irrigation Area experiences the wax yellow and harvest stages, respectively, and the water shortage has no impact on the harvest at this time.	This month is the harvest time for summer corn, so the water source situation has no impacts on the early drought warning.
October	The early drought warning situation is ideal, but the current drought situation needs to be focused on, which may be caused by low crops’ water demand; so administrators can carry out a drought prevention based on specific observations in the Irrigation Area.	Summer corn has been harvested when the autumn sowing preparation stage comes. So the soil moisture during this period only affects the winter wheat sowing.	This month is the sowing time for winter wheat, and the water source situation has no impacts on early drought warning.
November	Here, the drought situation and water source situation are all quite ideal because the crops’ water demand is not very high. However, with the declining of the temperature, administrators should consider the water requirements for the crops to live through the winter, so water storage in deep soil layers can be carried out.	As water irrigation based on stubble repeating for winter wheat is carried out, the soil water content is sufficient, satisfying the current growth needs for the winter wheat. But water storage in deep soil layers is insufficient, so water supplement in good time is suggested to lay a good foundation for the harvest in the next year.	This month is a preparation stage for winter wheat to grow through the winter.
December	The drought’s potential development trend appears in this month. When Q (the future water source situation) is greater than Q_50_, the signal shows the “blue light”, possibly having something to do with low temperature and in-surface water freezing. Administrators can carry out a deep soil irrigation to relieve the potential drought trend.	The minimum temperature has fallen below 0° for a few recent consecutive days, and the surface and its night field layer have begun to freeze, but because of the clear weather during day times, the 40 cm soil water in the winter wheat’s planning layer in the Irrigation Area is quite proper; however, it is a good opportunity for wheat to be irrigated in winter when the water storage is enough in the deeper soil layers.	Winter wheat enters the cultivating stage in this month; besides the planning layer of 40 cm in the Irrigation Area, the storage of water in the deeper soil layers should also be considered to make preparations for the harvest in the next year.

**Table 15 ijerph-17-00374-t015:** Water supply analysis in the Irrigation Area in three future months from March 2013 (T = 3).

Future Water Source Situation	t = 1 (April)	t = 2 (May)	t = 3 (June)
	*DAI*	*DAI*	*DAI*
Q_5_	0.86	1.00	0.43
Q_10_	0.86	1.86	1.29
Q_20_	0.86	1.86	1.29
Q_30_	0.86	1.86	1.29
Q_40_	1.72	3.00	1.80
Q_50_	1.72	3.00	1.80
Q_60_	2.00	3.00	1.80
Q_70_	2.00	3.00	1.80
Q_80_	2.00	3.00	2.15
Q_90_	2.00	3.00	2.15
Q_95_	2.00	3.00	2.43
Weight (Wt.)	0.40	0.33	0.27

**Table 16 ijerph-17-00374-t016:** Water supply analysis in the Irrigation Area for the three future months from September 2013 (T = 3).

Future Water Source Situation	t = 1 (October)	t = 2 (November)	t = 3 (December)
	*DAI*	*DAI*	*DAI*
Q_5_	0.86	0	0.43
Q_10_	0.86	0	0.86
Q_20_	0.86	0	0.86
Q_30_	0.86	0	0.86
Q_40_	0.86	0	0.86
Q_50_	0.86	0	0.86
Q_60_	0.86	0	1.36
Q_70_	0.86	0	1.36
Q_80_	0.86	0	1.36
Q_90_	0.86	0	1.72
Q_95_	0.86	0	1.72
Weight (Wt)	0.40	0.33	0.27

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
