# Peer review of "Evaluation on Early Drought Warning System in the Jinghui Channel Irrigation Area"

_ijerph, 2020, doi:10.3390/ijerph17010374_

Round 1

Reviewer 1 Report

The English of the paper has to be improved

The lines were numbered for each page and connected which makes it difficult for the reviewer.

The title is quite vague and should be more precise (may be remove"in changing environment and replace it with the name of the study region").
L20 change The to the
The region of study should be clearly mentioned in the abstract
L25 where is 5.1?
soil moisture at 40 cm or 1 m? please rewritten this part.

L16 What are these analyses situations? please add more information.

All values in the tables: are these spatial mean over the region?

Adding a map of the study region is a must.

The computation of D and S was not clear and the criteria that was used in the analyses was not clear enough

What is the purpose of table 5?

This reviewer thinks that runoff has important role but the authors did not consider it.

In table8, how K1,K2, etc a1 and a2 were identified? please provide real justifications.

3.6.1   this section is written like an exercise for students: do this and that then calculate...... please rewrite this paragraph, and through the paere, in a scitfilcly manner.
The methodology in general was not clear enough. the authors can add a diagram to explain the methodology. To better understand these equation, the authors may provide the scripts (codes) used to for the computations.

Table 14 can go to supplementary

For figures 3 and 5, there are big differences between actual warning and the drought levels!
Do the authors think this will convince decision makers to follow the  proposed methodology? a paragraph needs to be added tackling this issue!

Author Response

Reviewer 1

Comments and Suggestions for Authors

The English of the paper has to be improved

The English language has been further modified by experts to basically achieve the habit of English expression.

The lines were numbered for each page and connected which makes it difficult for the reviewer.

Modified.

The title is quite vague and should be more precise (may be remove"in changing environment and replace it with the name of the study region").

Modified.
L20 change The to the

Modified.
The region of study should be clearly mentioned in the abstract

The study area has been introduced in the revised abstract.

 L25 where is 5.1?

Modified. It should be formula (1).

soil moisture at 40 cm or 1 m? please rewritten this part.

It has been rewritten.

All values in the tables: are these spatial mean over the region?

Yes, they are spatial mean values.

 Adding a map of the study region is a must.

The study region map has been added in the ms.

The computation of D and S was not clear and the criteria that was used in the analyses was not clear enough

Determination of Drought index (D) is obtained based on the research method in reference [38] and [39],the calculation steps are as followings:

This paper has introduced the process of establishing and evaluating the drought index (D) of irrigation areas by using the fuzzy synthetic evaluation method [38-39]. This paper mainly has selected two influencing factors which are precipitation and soil moisture to establish the drought-evaluation index (D) in the form of weighted coupling, with particulars given as follows:

(1) Precipitation

In regards to the arid and semi-arid regions, the drought occurrence possibility is mainly dependent on precipitation in the meteorological conditions, so the precipitation index is taken as an evaluation one. In the early drought warning, the standardized precipitation index (SPI) is simpler than the Palmer drought severity index and is well-adapted, so it is used to represent meteorological drought in the irrigation area.

(2) Soil moisture

With respects to the Jinghui Channel Irrigation Area, the soil moisture is a direct drought representation, so its impacts are taken into consideration during drought evaluation. Also its statistics are picked out from soil moisture data which are found in the official monitoring results provided by Jinghui Channel Management Office.

(3) Classification of Drought grades

The classification of drought grades should fully reflect the drought changes scope in the research area. According to provisions in the Compilation Guidelines for Drought Emergency Plan, the drought situations in the Jinghui Channel Irrigation Area are classified into the following five grades: No drought (V1), mild drought (V2), moderate drought (V2), serious drought (V4) and extreme drought (V5).

Water Source Situation Index (S) Determination is based on references [40], [41] and [42] the process of which was emitted here.

What is the purpose of table 5?

Assuming that the inflow is changed from Q5 to Q95 in future T0 time periods, then the description of early warning index of the different inflow probability is given as shown in Table 5.

This reviewer thinks that runoff has important role but the authors did not consider it.

In this paper, we consider that runoff is related to the precipitation and soil moisture. The runoff will infiltrate into the soil when the area is dry and the rainfall intensity is large enough.

In table8, how K1,K2, etc a1 and a2 were identified? please provide real justifications. Determination of the membership function levels defined through the triangular fuzzy distribution method.

3.6.1   this section is written like an exercise for students: do this and that then calculate...... please rewrite this paragraph, and through the paere, in a scitfilcly manner.

Modified.

The methodology in general was not clear enough. the authors can add a diagram to explain the methodology. To better understand these equation, the authors may provide the scripts (codes) used to for the computations.

In this section, we first collected the data of drought index (D) and future water situation (S) in 2013 to calculate the drought early warning under different water source condition. The warning signals were illustrated in table 13. Based on the process of drought warning, the drought early warning is conducted combined drought situation and future water resources index (D) and (S). Then, compare the early drought warning simulation results with the Jinghui Channel Irrigation Area’s monitoring ones released by officials. The comparison results are illustrated as follows: (1) Meteorological drought grades and actual early warning signals comparison ï¼›(2) Soil moisture - related drought degrees and actual early warning signals comparison ï¼›(3) Drought grades (D) and actual early warning signals comparison.

Table 14 can go to supplementary

Considering the continuity of the tables, Table 14 is still kept in the ms.

For figures 3 and 5, there are big differences between actual warning and the drought levels!

the actual early warning signals are not consistent with precipitation-related drought degrees, and for the Jinghui Channel Irrigation Area, the precipitation amount explains a fundamental reason for the drought, but it is not the only decisive factor. Only when the relationship between the soil moisture and the balance between supply and demand is also considered can the realistic information about early drought warning be obtained.

For the Jinghui Channel Irrigation Area, the soil moisture is the most direct manifestation of the drought degrees, so its variation trend will definitely influence that of the early drought warning, but both the precipitation and the relationship between the supply and demand also impact the early drought warning results.

Do the authors think this will convince decision makers to follow the  proposed methodology? a paragraph needs to be added tackling this issue!

We do think this methodology could be a very useful tool for management officer to conduct and make some decisions. With the measured data of precipitation and soil moisture, the early warning signals could be obtained ahead of the real drought occurring. This method has been applied in the Jinghui channel irrigation area for an example. And it could be conducted in other regions which also face severe drought situations in the specific season under the global changing climate.

Reviewer 2 Report

Evaluation on Early Drought Warning System in Irrigation Areas in a Changing Environment

General comments

This paper has aimed to make a basic framework of the early drought warning system.

For this reason, they select two factors including precipitation and soil moisture in order to establish the drought- evaluation index (D) in the form of weighted coupling, and establish the “green, blue, yellow, orange and red lights” as the early warning grades for agricultural drought through the determination of the water source situation index (S) to reflect a comprehensive index value concerning the disaster crisis that the irrigation area may face in the future. Unfortunately, there is no uniqueness in this study and the methodology, results and conclusions do not provide new and original developments on the investigated subject. Therefore, the reviewer recommends this manuscript with major revisions before publication.

Specific comments

Please provide novelty of your study? Which new information we can obtain by your study regardless of specific application of the results? There are too much explanations regarding early drought warning definition and meaning which should be summarized. Please provide monthly information about drought warning regarding in which month the drought warning is green, blue, yellow, orange and red lights. Please provide a discussion part for your paper and compare your results with the pervious studies.

Author Response

This paper has aimed to make a basic framework of the early drought warning system.

For this reason, they select two factors including precipitation and soil moisture in order to establish the drought- evaluation index (D) in the form of weighted coupling, and establish the “green, blue, yellow, orange and red lights” as the early warning grades for agricultural drought through the determination of the water source situation index (S) to reflect a comprehensive index value concerning the disaster crisis that the irrigation area may face in the future. Unfortunately, there is no uniqueness in this study and the methodology, results and conclusions do not provide new and original developments on the investigated subject. Therefore, the reviewer recommends this manuscript with major revisions before publication.

Specific comments

Please provide novelty of your study? Which new information we can obtain by your study regardless of specific application of the results? There are too much explanations regarding early drought warning definition and meaning which should be summarized. Please provide monthly information about drought warning regarding in which month the drought warning is green, blue, yellow, orange and red lights. Please provide a discussion part for your paper and compare your results with the pervious studies.

This research has innovative results in the drought early warning, which aims to construct the basic framework of drought early warning system. Based on the theory of drought early warning, we established "green, blue, yellow, orange and red lights" for agricultural drought disaster warning level (The theory is first proposed by Prof. Wenzheng Huang of Taiwan Ocean University. The author of this paper is one of the main researchers during the visit of the Taiwan Ocean University.).

We selected two main factors of precipitation and soil entropy to construct the evaluation model of drought assessment index using weighted coupling method. Then, this method was applied for Jinghui canal irrigation area.

The innovative points of this paper: (1) the drought early warning system of irrigation area is constructed by using "Green, Blue, Yellow, Orange, and Red lights" for the early warning grade of agricultural drought disaster. The research structure and framework is one of the important models to deal with this kind of problem, which can be used for reference to investigate the research structure of drought early warning. (2) the specific situation from January to December in Jinghui Canal Irrigation District is obtained. The actual drought early warning shows a "green light" due to the ideal soil moisture in January and February; In March, the soil water shortage was serious, but the incoming water situation was ideal, so the actual drought early warning was "green light". From April to August, the actual early warning lights are basically the same as the drought degree of soil moisture. After September, the actual early warning is "green light", but the soil moisture is not very ideal, which is due to the harvest time of summer corn and the sowing date of winter wheat in September and October. The lack of soil moisture has no effect on the yield of crops, and the demand for water in the growth and development stage of winter wheat in November and December is not very great. (3) This warning system has certain practical guiding significance for drought prevention in this area. Meanwhile, the theoretical framework of this study also has great significance for the studies in other areas.

Reviewer 3 Report

Its a normal paper on EWD.

Author Response

Thank you for your review of this article. We thank the reviewers for their recognition of this article. The author will continue to study the problem in depth.

Reviewer 4 Report

In my opinion, the article does not meet the standards of a scientific article. Individual chapters give the impression of being written separately, with no common thread.

Comments:

- lack of proper introduction to the literature on the problem,

- The following information is missing: description of the study area, maps with the location of the study area, physical and geographical conditions (elevation of the area, geological structure, river network etc.)

- Chapter 2 is a total chaos. The methodology is not clearly described. There is no information about where the rainfall data comes from, what is the supply and demand for water. It is not clearly explained why, for the assessment of drought, the authors take the SPI precipitation index, and in case of soil humidity - actual raw values. There is no explanation for this approach. Lack of information on water supply and demand. The authors cite the research of Professor Huang Wenzheng without quoting literature. Some formulas have been simplified without proper explanation of the assumptions.

- Chapter 3 contains several minor grammatical errors, but is more structured. There is no explanation as to: why 2013 was adopted for analysis and why certain crops are mentioned, since the introduction and methodology have no mention about it.

- Conclusions are not fully supported by the results

Author Response

In my opinion, the article does not meet the standards of a scientific article. Individual chapters give the impression of being written separately, with no common thread.

Comments:
- lack of proper introduction to the literature on the problem,
The revised version has added literature on the problem in introduction.
- The following information is missing: description of the study area, maps with the location of the study area, physical and geographical conditions (elevation of the area, geological structure, river network etc.)
Jinghui canal: the zheng guo canal built during the warring states period in 246 bc. after the drought in guanzhong in the late 1920s, the irrigation project of jinghui canal was built with a planned irrigation area of 640,000 mu at that time, and a real irrigation area of 500,000 mu by 1949. Since then, the canal head project and irrigation and drainage system have been expanded and rebuilt. The irrigation area is irrigated by a single source of water and developed into a multi-source irrigation system combined with canal wells. Taking Jinghui Canal as the center, connecting Baoji Gorge, Wool Bay, Fengjiashan and other irrigation areas in the west, Dongfanghong, Luohui Canal and Lei Yanghuang irrigation areas in the east, in Baoji in the west and Weibei Plateau in the hundreds of miles from the Yellow River in the east, a river canal is formed to form a network, the surface water is combined with groundwater, and the large-scale irrigation system of nearly 670,000 hectares of the tank land is turned into the main grain production base in Shaanxi Province.

- Chapter 2 is a total chaos. The methodology is not clearly described. There is no information about where the rainfall data comes from, what is the supply and demand for water. It is not clearly explained why, for the assessment of drought, the authors take the SPI precipitation index, and in case of soil humidity - actual raw values. There is no explanation for this approach. Lack of information on water supply and demand. The authors cite the research of Professor Huang Wenzheng without quoting literature. Some formulas have been simplified without proper explanation of the assumptions.
The revised version of chapter 2 has analyzed in the following order: (1) the detailed information of the study area; (2) introduction of the analytical workframe and research method of the drought index, including ①Determination of Drought index (D);②Water Source Situation Index (S) Determination;③Establishment of Early Drought Warning Framework System. (3) investigate and evaluate the drought level. The revised version has added the research article of Wenzheng Huang in the reference section.

- Chapter 3 contains several minor grammatical errors, but is more structured. There is no explanation as to: why 2013 was adopted for analysis and why certain crops are mentioned, since the introduction and methodology have no mention about it.
This part has been rewritten.
The data was adopted because the drought situation was relatively severe in northwest region of 2013. The water supply project is not complete enough, and the supply of artificial water supply resources is insufficient.
- Conclusions are not fully supported by the results
The author has revised the conclusion and added summary to support the results. The soil entropy is the direct factor which influences the drought of irrigation area. It has great effects on drought warning but not the unique factor. The drought index (D) is a comprehensive indicator which combined precipitation and soil entropy. Meanwhile, it is related to drought warning. The drought situation in this research performed consistency in January, February, May, July, November and December. There exists differences in other months, because the water diversion at the head of canal and groundwater exploitation were considered which reflects the regulating role of human activities in the process of drought early warning.

Round 2

Reviewer 1 Report

All my comments were addressed.I have no more comments.

Author Response

Dear reviewer,

Thank you for your acceptance of this article.

Best regards

Pro.Shibao Lu

Reviewer 2 Report

I have reviewed the revised manuscript and I believe that it has satisfactorily addressed the issues. Therefore, I am happy to accept your paper for publication in the journal.

Author Response

(The authors gave the same response as above.)

Reviewer 4 Report

In my opinion, the article does not meet the standards of a scientific article. In my opinion the paper is not suitable for publication.
1. Comments from the previous review have not been taken into account
2. The updates contain errors

- references are listed incorrectly e.g.
Page 25. line 43 "Souza, Jovani Taveira, etc. (2019). Sustainable development and economic performance: Gaps and trends for future research. SUSTAINABLE DEVELOPMENT .DOI: 10.1002 / sd.1982 "
page 26 line 21 "Qian, QK etc. (2019). Does aging-friendly enhance sustainability? Evidence from Hong Kong. SUSTAINABLE DEVELOPMENT, 2019,27: 657-668 "
Page 26 line 31 "Madreimov etc. (2019). Natural-resource dependence and life expectancy: A nonlinear relationship. SUSTAINABLE DEVELOPMENT, 2019,27: 681-69 "
- the text contains comments that should not be included in the article, e.g. page 24, line 26 "(4) The author has modified the conclusion section which had added a summary that supported the results."

Author Response

Dear reviewer4,

Thanks to the reviewers for their careful evaluation. the quality of this paper has been greatly improved through the comments of the reviewers. The authors made further modifications based on the comments of the reviewers.

Best regards

Pro. Shibao Lu